# Evaluation of Performance of Three Satellite-Derived Precipitation Products in Capturing Extreme Precipitation Events over Beijing, China

**Yu Li** [1,2], **Bo Pang** [1,2,*], **Meifang Ren** [3], **Shulan Shi** [1,2], **Dingzhi Peng** [1,2], **Zhongfan Zhu** [1,2] and **Depeng Zuo** [1,2]

1    College of Water Sciences, Beijing Normal University, Beijing 100875, China;
202021470013@mail.bnu.edu.cn (Y.L.); 201821470021@mail.bnu.edu.cn (S.S.); dzpeng@bnu.edu.cn (D.P.);
zhuzhongfan1985@bnu.edu.cn (Z.Z.); dpzuo@bnu.edu.cn (D.Z.)
2    Beijing Key Laboratory of Urban Hydrological Cycle and Sponge City Technology, Beijing 100875, China
3    China Academy of Urban Planning and Design, Beijing 100037, China; renmeifang@mail.bnu.edu.cn
*    Correspondence: pb@bnu.edu.cn; Tel.: +86-135-2168-6445

**Abstract:** Extreme precipitation events have a more serious impact on densely populated cities and therefore reliable estimation of extreme precipitation is very important. Satellite-derived precipitation products provide precipitation datasets with high spatiotemporal resolution. For improved applicability to estimating urban extreme precipitation, the performance of such products must be evaluated regionally. This study evaluated three satellite-derived precipitation products, the Integrated Multi-satellite Retrievals for GPM (IMERG_V06), Multi-Source Weighted-Ensemble Precipitation (MSWEP V2), and China Meteorological Forcing Dataset (CMFD), in capturing extreme precipitation using observations acquired at 36 rainfall stations during 2001–2016 in Beijing, China. Results showed that MSWEP had the highest accuracy regarding daily precipitation data, with the highest correlation coefficient and the lowest absolute deviation between MSWEP and the rainfall station observations. CMFD demonstrated the best ability for correct detection of daily precipitation events, while MSWEP maintained the lowest rate of detecting non-rainy days as rainy days. MSWEP performed better in estimating precipitation amount and the number of precipitation days when daily precipitation was <50 mm; CMFD performed better when daily precipitation was >50 mm. All three products underestimated extreme precipitation. The Structural Similarity Index, which is a map comparison technique, was used to compare the similarities between the three products and rainfall station observations of two extreme rainstorms: "7.21" in 2012 and "7.20" in 2016. MSWEP and CMFD showed higher levels of similarity in terms of spatial–temporal structure. Overall, despite systematic underestimation, MSWEP performed better than IMERG and CMFD in estimating extreme precipitation in Beijing.

**Keywords:** satellite-derived precipitation products; IMERG; MSWEP; CMFD; extreme precipitation

## 1. Introduction

Urban extreme precipitation can be one of the most hazardous weather events owing to the concentration of population and wealth in urban areas [1]. Extensive efforts have been dedicated to studying urban extreme precipitation [2,3]. Recent related studies have shown that urbanization might increase the frequency and intensity of regional extreme precipitation [4]. Therefore, detection of changes in urban extreme precipitation has been a critical research focus in recent decades [5,6].

The main methods adopted for monitoring urban precipitation include rainfall station observations, weather radar observations, and satellite-derived products [7]. Rainfall station observations represent the most direct and precise method for acquiring precipitation data. However, use of interpolation methods results in potential errors when obtaining continuous spatial precipitation estimates owing to the sparse and uneven distribution

of rainfall stations over most continents [8,9]. Radars measure precipitation indirectly by radiating electromagnetic energy and collecting the echo reflected by water droplets, which can produce an instantaneous distribution of precipitation over a wide range and provide real-time high-resolution precipitation estimates [10]. Thus, radars can compensate for the inadequacies of areal rainfall estimation based on rainfall station observations, and represent an effective tool for monitoring hazardous weather, quantitative estimation of precipitation, and quantitative precipitation forecasting. Nevertheless, weather radar observations of precipitation are often affected by factors such as complex terrain and cloud movement, which can introduce high levels of uncertainty [11]. Satellite-derived products, which can provide high spatiotemporal resolution precipitation information with large coverage, represent an alternative to rainfall station observations, especially for cities with limited numbers of rainfall stations [12]. Various satellite-derived products have been produced and applied to monitor urban precipitation, e.g., Tropical Rainfall Measuring Mission (TRMM) [13–15], Climate Prediction Center morphing technique (CMORPH) [16–18], Precipitation Estimation from Remotely Sensed Information using Artificial Neural Networks (PERSIANN) [19–21], and the Integrated Multi-Satellite Retrievals for GPM (IMERG) [22–24].

The accuracy of satellite-derived precipitation products is affected by errors in sensor observations and retrieval algorithms, especially regarding extreme values [25]. Therefore, it is of great importance to evaluate the precision and uncertainties of such products in capturing extreme values of precipitation before application to a specific region [26,27]. Numerous previous studies used extreme precipitation indices and categorical error metrics to evaluate the performance of satellite-derived precipitation products in estimating extreme precipitation and detection skill in different regions [13,28–31]. Some studies analyzed the performance of the satellite-derived precipitation products in relation to specific extreme precipitation events, including evaluating the capability of satellite-derived precipitation products in quantifying typhoon-related extreme rainfall [32,33] and extreme precipitation events in different time series [34]. Most of these studies adopted a series of indices and metrics to compare the differences between satellite-derived precipitation products and rainfall station observations. However, similarities in the spatial structure of extreme precipitation events, which are essential in relation to metropolitan areas with pronounced spatial differences in population and poverty, have rarely been studied. The Structural Similarity Index (SSI), which was developed to quantify the similarity between a compressed image and a reference image [35], has shown applicability to comparison of spatial ecological data [36,37] and annual precipitation distribution [38]. Theoretically, SSI is highly suitable for analysis of similarities in the spatial structure of extreme precipitation events.

Many satellite-derived precipitation products have been evaluated and applied in the field of hydrometeorology [39–41]. Among these products, the Integrated Multi-Satellite Retrievals for GPM (IMERG_V06), Multi-Source Weighted-Ensemble Precipitation, version 2 (MSWEP V2), and China Meteorological Forcing Dataset (CMFD), have shown strong applicability to monitor extreme precipitation. The newly released IMERG product provides global estimates of precipitation with high temporal and spatial resolutions. Some studies have found that IMERG has greater estimation accuracy in detecting moderate rain, heavy rain, and rainstorms in low-elevation areas along the eastern coast of China [42,43], with reports that GPM/IMERG outperforms The TRMM Multi-satellite Precipitation Analysis (TMPA)products in representing the spatial pattern, overall volume, and probability characteristics of extreme precipitation over China [27,44]. MSWEP V2.2 is a gridded dataset with high spatiotemporal resolution and a long time series, which integrates site data, satellite data, and reanalysis data, as well as corrections based on runoff and potential evapotranspiration data. Liu et al. [45] evaluated the accuracy of Climate Hazards Group Infrared Precipitation with Station data Version2 (CHIRPS v2) [46] and MSWEP V2 and found that MSWEP has higher accuracy than CHIRPS in relation to extreme precipitation on the Qinghai–Tibet Plateau. CMFD is a gridded near-surface meteorological dataset with high spatiotemporal resolution [47], which has been proven to perform better than both

Asian Precipitation-Highly Resolved Observational Data Integration Towards Evaluation of Water Resources (APHRODITE) [48] and CHIRPS in capturing the spatiotemporal pattern of the most extreme precipitation indices over the Qinghai–Tibet Plateau in China [49]. Most of these previous studies focused on global, national, and basin scales, while few studies have evaluated the performance of such datasets in monitoring extreme precipitation at the scale of a metropolitan area.

In this paper, comprehensive comparison is conducted to evaluate the performance of satellite-derived precipitation products in metropolitan areas, which is helpful for hydrological research to select appropriate precipitation products for the study of flood early warning. In addition to conventional extreme precipitation metrics on frequency, intensity, occurrence date, and capture ability, the SSI approach is adopted to compare the spatial structure of extreme rainstorms captured by satellite-derived precipitation products and rainfall station observations. The proposed method is adopted to evaluate the performance of IMERG, MSWEP, and CMFD over Beijing using daily precipitation data acquired at 36 rainfall stations during 2001–2016. The remainder of this paper is organized as follows. In Section 2, the study area and the precipitation datasets used are introduced. In Section 3, we briefly introduce the research methods and selected indicators. The derived results and their analysis are presented in Section 4. In Section 5, we discuss these results, and Section 6 presents our conclusions.

## 2. Study Area and Data

### 2.1. Study Area

Beijing, which is located in the northern part of the North China Plain (39.4°–41.6°N, 115.7°–117.4°E), covers an area of 16,410.54 km$^2$ [50]. Generally, the elevation of the terrain is high in the northwest and low in the southeast. The mountainous area accounts for approximately 62% of the total area, and plains account for approximately 38% [51]. Beijing has a subhumid continental monsoon climate of the northern temperate zone, with hot rainy summers, cold dry winters, and uneven distribution of precipitation [52]. Approximately 80% of the annual precipitation is concentrated in summer (June–August). According to statistics from the Beijing Water Resources Bulletin, the average rainfall in Beijing (1950–2012) is 585 mm. With continuous urbanization, the permanent population of Beijing has increased from 13.64 million in 2000 to 21.54 million in 2018. According to previous studies, urbanization and topography have influenced the frequency and intensity of extreme precipitation in Beijing [4,53,54]. Referring to these studies, we divided Beijing into the following six subregions: the urban area (UA), inner suburban area in the south (ISAS), inner suburban area in the north (ISAN), outer suburban area (OSA), southwest mountainous area (SWMA), and northwest mountainous area (NWMA) (Figure 1) [55].

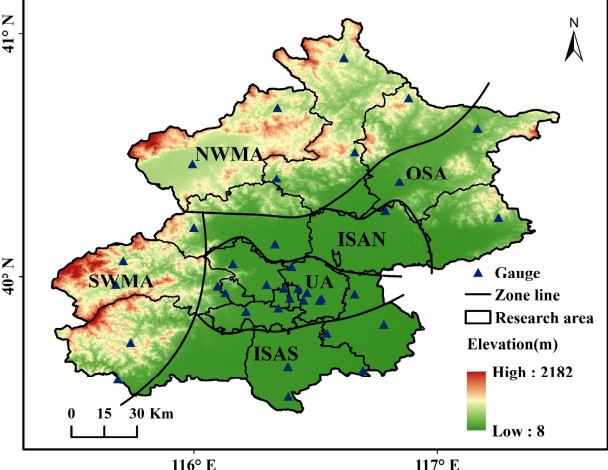

**Figure 1.** Locations of the rainfall stations in the six subregions of Beijing superimposed over a topographic map.

### 2.2. Datasets

2.2.1. Satellite-Derived Precipitation Products

IMERG is a level 3 product of the Global Precipitation Measurement (GPM) mission. It makes full use of all the data provided by the satellite sensors onboard the GPM platform and it also borrows from the previous TRMM era. IMERG uses passive microwave data from recent instruments, including Defense Meteorological Satellite Program Flight 19 (DMSP-F19, developed by Lockheed Martin Space Systems Company, Rockville, MD, USA), GPM Microwave Imager (GMI, developed by Ball Aerospace and Technology Corporation, Boulder, CO, USA), and National Oceanic and Atmospheric Administration–20 (NOAA-20, developed by the National Oceanic and Atmospheric Administration, Boulder, CO, USA), which provide reasonably accurate satellite-based precipitation estimates. IMERG uses data from as many low Earth orbit satellites as possible to compensate for the limited sampling available from single low Earth orbit satellites, and to augment infrared precipitation estimates from geosynchronous Earth orbit satellites [56]. Previous studies have proven that the accuracy of IMERG products is notably improved in comparison with that of TRMM products [57,58], and the IMERG products have proved useful in studying climate change and conducting hydrological simulations [59]. Since 2014, IMERG has released four official versions of data, and the previous version was replaced by IMERG_V06 in 2019. According to different data processing procedures, IMERG provides three types of precipitation product: the near-real-time Early-Run and Late-Run products and the delayed Final-Run product [60]. This study used the IMERG_ V06 Early-Run precipitation dataset (2001–2021), which has spatial and temporal resolutions of 0.1° and 1 d, respectively.

MSWEP V2 is a global precipitation dataset recently developed by Beck et al. [61]. The spatial and temporal resolutions of MSWEP are 0.1° and 3 h, respectively, and the data series length covers 1979–2017. In comparison with version 1, MSWEP V2 supplements reanalysis and rainfall station data with rainfall estimates based on infrared data to improve the precipitation estimates in convection-dominated regions. The product integrates weather station data (e.g., global daily climate history network data and global ground daily weather data), satellite observation data, including CMORPH, Gridded Satellite (GridSat), The Global Satellite Mapping of Precipitation (GSMaP), and TRMM 3B42RT, and model simulation data, including Interim ECMWF Re-Analysis (ERA-Interim) and the Japanese 55-year Reanalysis (JRA-55). In some watersheds, calibration is performed using runoff and potential evapotranspiration data, which have the characteristics of long time series and high spatial resolution [62]. Beck et al. [61] also compared the performance of 11 types of global or quasi-global precipitation datasets (including 1 pure measurement dataset and 10 combined datasets) on the daily scale using rainfall station and radar precipitation data in the United States and found that MSWEP V2 always had the highest accuracy. In this study, the daily precipitation dataset of MSWEP V2(1979–2016) was obtained by accumulating 3 h observations of precipitation.

CMFD [47] is a near-surface meteorological and environmental reanalysis dataset developed by the Institute of Tibetan Plateau Research of the Chinese Academy of Sciences, which includes near-surface air temperature, near-surface air pressure, near-surface specific humidity, near-surface wind speed, surface downward shortwave radiation, surface downward longwave radiation, and surface precipitation rate. The spatial and temporal resolutions of CMFD are 0.1° and 3 h, respectively. The data series length covers 1979–2018. The dataset is based on internationally produced Princeton reanalysis data, Global Land Data Assimilation System data, the Global Energy and Water Cycle Exchanges Surface Radiation Budget (GEWEX-SRB) radiation data, and TMPA 3B42 calibration products as the background field, and integrated with China's meteorological observation data. The accuracy is between that of meteorological station observations and satellite-derived data. In this study, the daily precipitation dataset of CMFD was obtained by accumulating 3 h observations of precipitation.

For comparison with the rainfall station data, the spatial and temporal resolutions of the three precipitation products were set at 0.1° and daily, respectively. Three satellite-derived precipitation products are demonstrated in Table 1.

**Table 1.** Summary information for the three satellite-derived precipitation products used in this study.

| Satellite Product | Temporal Resolution | Space Resolution | Temporal Span | Source Data | References |
|---|---|---|---|---|---|
| IMERG V06 Early-Run | Daily | 0.1° | 2001–2021 | Satellite data | [58] |
| MSWEP V2 | 3 h | 0.1° | 1979–2016 | Gauge, satellite, reanalysis data | [61] |
| CMFD | 3 h | 0.1° | 1979–2018 | Gauge, satellite, reanalysis data | [47] |

### 2.2.2. Rainfall Station Data

The precipitation observation data selected for use in this study were derived from daily precipitation data (2001–2016) acquired at 36 rainfall stations in Beijing and provided by the Beijing Municipal Hydrological Bureau and the China National Climate Center. The quality of the dataset is strictly controlled before its release. Except for the dense distribution of rainfall stations in urban areas of Beijing, the distribution of rainfall stations in suburban and mountainous areas is sparse and uneven. There are 15 rainfall stations in the urban area, 5 and 2 rainfall stations in the in the southern and northern suburbs, and 3 rainfall stations in the northern outer suburbs. Rainfall stations in mountainous areas are sparsely distributed, with 5 and 6 rainfall stations in the southern and northern mountains areas, respectively. The distribution of rainfall stations is shown in Figure 1.

### 3. Methods

We divided the comparison indices of extreme precipitation into the quantitative index, classification scoring index, extreme precipitation index, and SSI. Generally, there are two approaches, point-to-pixel and pixel-to-pixel methods, that can be applied to compare the satellite-derived precipitation products with rainfall station observations. The pixel-to-pixel method requires spatial interpolation of rainfall station observations to match the pixel-based data of satellite-derived products, which might create uncertainties, particularly owing to the sparse and uneven distribution of the rainfall stations [30,63], so we used a point-to-pixel approach to compare the rainfall stations data and the satellite-derived precipitation products data. This methodology has been used widely in assessing precipitation estimated by satellite-derived precipitation products [32,64] and it is the closest matching method in this study for the uneven distribution of rainfall stations in Beijing. We calculated the regional metrics by averaging the metrics of the rainfall stations in each subregion.

### 3.1. Quantitative Index

To compare and evaluate the accuracy of IMERG, MSWEP, and CMFD data in the Beijing area, this study adopted four statistical indictors: absolute deviation (AD), relative deviation (RB), root mean square error (RMSE), and correlation coefficient (Corr). We calculated indictor values corresponding to each rainfall station, and spatially averaged the indictor values over different subregions. These indicators were used to analyze both the spatial characteristics of the errors of the satellite-derived precipitation data at different sites in Beijing, and the errors generated by satellite-derived precipitation products under different levels of precipitation intensity. In addition, these indicators are also used to evaluate the accuracy of the occurrence dates of extreme precipitation detection by the rainfall stations and satellite-derived precipitation products. Their perfect values are

unity for Corr, and zero for AD, RB, and RMSE. The specific calculation methods are as follows [65,66]:

$$AD = \frac{\sum_{i=1}^{n} |S_i - G_i|}{n}, \tag{1}$$

$$RB = \frac{\sum_{i=1}^{n} (S_i - G_i)}{\sum_{i=1}^{n} (G_i)} * 100\%, \tag{2}$$

$$RMSE = \sqrt{\frac{\sum_{i=1}^{n} (S_i - G_i)^2}{n}}, \tag{3}$$

$$Corr = \frac{\sum_{i=1}^{n} (S_i - \overline{S})(G_i - \overline{G})}{\sqrt{\sum_{i=1}^{n} (S_i - \overline{S})^2} \sqrt{\sum_{i=1}^{n} (G_i - \overline{G})^2}}, \tag{4}$$

where $n$ is the number of samples; $S$ and $G$ are the satellite-derived products data and the rainfall station observations data, respectively; $\overline{S}$ is the average of the satellite-derived products data; and $\overline{G}$ is the average of the observation data of the ground stations.

*3.2. Classification Scoring Index*

To evaluate the ability of IMERG, MSWEP, and CMFD data to capture precipitation events, the threshold of precipitation occurrence was set at 0.1 mm/d, and two probability statistical indicators were selected: the probability of detection (POD) and the false alarm rate (FAR). Moreover, to compare and analyze the proportion of precipitation data successfully detected and falsely reported by satellite precipitation data, two volume statistical indicators were selected: the volumetric POD (VPOD) and the volumetric FAR (VFAR). The critical success index (CSI) [67,68] was adopted to analyze whether the precipitation events captured by the three satellite-derived precipitation products were consistent with the precipitation events observed by the ground stations, and to evaluate quantitatively the ability of the satellite products to capture precipitation events and non-precipitation events. The closer that POD, VPOD, and CSI are to 1, and the closer the FAR and VFAR are to 0, the more accurate the satellite-derived precipitation products are at capturing precipitation events. Each pair of daily precipitation records from the meteorological stations and corresponding grids of IMERG, MSWEP, and CMFD was classified as a "hit", "miss", or "false". The classification results are shown in Table 2.

**Table 2.** Contingency table of the satellite-derived precipitation data and rainfall station data.

| | Rainfall Stations ≥0.1 mm/d | Rainfall Stations <0.1 mm/d |
|---|---|---|
| Satellite-derived data ≥ 0.1 mm/d | Hit | False |
| Satellite-derived data < 0.1 mm/d | Miss | 0 |

The calculation methods of the indicators are as follows:

$$POD = \frac{N_{hit}}{N_{hit} + N_{miss}}, \tag{5}$$

$$FAR = \frac{N_{false}}{N_{hit} + N_{false}}, \tag{6}$$

$$CSI = \frac{N_{hit}}{N_{hit} + N_{false} + N_{miss}}, \tag{7}$$

$$VPOD = \frac{\sum_{i=1}^{n} [S_i | (S_i \geq thr \& G_i \geq thr)]}{\sum_{i=1}^{n} [S_i | (S_i \geq thr \& G_i \geq thr)] + \sum_{i=1}^{n} [S_i | (S_i < thr \& G_i \geq thr)]}, \tag{8}$$

$$VFAR = \frac{\sum_{i=1}^{n} [S_i | (S_i \geq thr \& G_i < thr)]}{\sum_{i=1}^{n} [S_i | (S_i \geq thr \& G_i \geq thr)] + \sum_{i=1}^{n} [S_i | (S_i \geq thr \& G_i < thr)]}, \tag{9}$$

where $n$ is the number of samples, $N_{hit}$ is the total count of days when both rainfall stations and satellite-derived products observe rain, $N_{false}$ is the total count of days when the satellite-derived products detect rain that is not detected by the rainfall stations, $N_{miss}$ is the total count of days when the satellite-derived products miss rain that is detected by the rainfall stations, and *thr* is the precipitation threshold. The index values obtained through different specifications of *thr* reflect the inversion accuracy of the satellite-derived data for different levels of precipitation.

To evaluate the capability of the satellite-derived precipitation products to capture extreme precipitation, we first calculated the annual precipitation for each station at thresholds of the 75th, 80th, 85th, 90th, 95th, and 99th quantiles, and the numbers of days for which the precipitation data measured by the stations were greater than the thresholds. The values of the rainfall stations were averaged in the research area, as shown in Table 3. Precipitation above the thresholds of the 75th and 90th quantiles was selected for evaluation to ensure sufficient precipitation events for calculation and to reflect the differences in precipitation capture capability of the different thresholds. We calculated the POD and FAR of the three products for precipitation events when the measured values of the rainfall stations were above the thresholds of 75th and 90th quantiles (Q75 threshold and Q90 threshold).

**Table 3.** Rainfall stations with different thresholds of precipitation and the numbers of days when precipitation was > *thr*.

| | Thresholds | | | | | |
|---|---|---|---|---|---|---|
| | **R75p** | **R80p** | **R85p** | **R90p** | **R95p** | **R99p** |
| **Rainfall stations (mm)** | 10.0 | 12.7 | 16.6 | 22.8 | 34.4 | 73.1 |
| **Rdays (Precipitation > *thr*)** | 18 | 15 | 11 | 7 | 4 | 1 |

### 3.3. Extreme Precipitation Index

This study selected 13 widely used precipitation indicators, defined by the Expert Team on Climate Change Detection and Indices (ETCCDI) [69,70], to analyze the data accuracy of IMERG, MSWEP, and CMFD in terms of extreme precipitation. The definitions of these 13 extreme precipitation indices are presented in Table 4. The annual total precipitation (ATP) and annual mean precipitation intensity (API) are equivalent to annual total wet-day precipitation (PRCPTOT) and Simple Daily Intensity Index (SDII), respectively. The consecutive wet day (CWD) and consecutive dry day (CDD) indicate the maximum number of wet or dry durations in a period of time. The RX1day and RX5day were selected as the max indices and the percentile indices were obtained from the 95th percentile of daily precipitation on wet days. Since the 99th percentile of daily precipitation in Beijing is basically equal to the RX1day, the 99th percentiles were excluded. The Rnnmm index represents the number of precipitation days in different precipitation levels based on the classification of precipitation levels by China Meteorological Administration and the characteristics of daily precipitation in Beijing [71]. We calculated the annual 13 extreme precipitation indexes from 2001 to 2016 by using the annual daily precipitation series detected by the rainfall station and satellite-derived precipitation products.

**Table 4.** Detailed information regarding extreme precipitation indices.

| Sort | Index | Definition | Units |
|---|---|---|---|
| Total indices | ATP | Annual total precipitation | mm |
| | API | Annual mean precipitation intensity | mm/day |
| Persistent indices | CDD | Maximum number of consecutive dry days | days |
| | CWD | Maximum number of consecutive wet days | days |
| Max indices | RX1day | Annual max 1-day precipitation | mm |
| | RX5day | Annual max 5 days of consecutive precipitation | mm |
| Percentile indices | R95p | The 95th percentile of daily precipitation on wet days | mm |
| | R95pTOT | The annual sum of precipitation on days where daily precipitation exceeds the 95th percentile of daily precipitation | mm |
| Absolute threshold indices | R0.1 mm | Annual count of days when daily precipitation is between 0.1 and 5 mm | days |
| | R5 mm | Annual count of days when daily precipitation is between 5 and 10 mm | days |
| | R10 mm | Annual count of days when daily precipitation is between 10 and 25 mm | days |
| | R25 mm | Annual count of days when daily precipitation is between 25 and 50 mm | days |
| | R50 mm | Annual count of days when daily precipitation is >50 mm | days |

### 3.4. Structural Similarity Index

We adopted the SSI to quantify the spatial structure difference of two extreme rainstorms ("7.21" in 2012 and "7.20" in 2016) captured by the satellite-derived precipitation products and the rainfall stations. The SSI compares the values of each grid between satellite-derived precipitation products and the rainfall station data, while also accounting for neighboring grids' values. To account for the greater spatial differences between the satellite-derived precipitation products and the rainfall station data, we used the lower limit of neighborhood size ($3 \times 3$ cells) and performed quantitative analysis by estimating the similarity in mean (SIM), similarity in variance (SIV), and similarity in pattern (SIP) [35]. The equations for these metrics are as follows:

$$\text{SIM}(S, G) = \frac{2\mu_s\mu_g + c_1}{\mu_s^2 + \mu_g^2 + c_1}, \tag{10}$$

$$\text{SIV}(S, G) = \frac{2\sigma_s\sigma_g + c_2}{\sigma_s^2 + \sigma_g^2 + c_2}, \tag{11}$$

$$\text{SIP}(S, G) = \frac{2\sigma_{sg} + c_3}{\sigma_s\sigma_g + c_3}, \tag{12}$$

$$u_s = \sum_{i=1}^{n} w_i P_{si}, \tag{13}$$

$$\sigma_s^2 = \sum_{i=1}^{n} w_i (P_{si} - u_s)^2, \tag{14}$$

$$\sigma_{sg} = \sum_{i=1}^{n} w_i (P_{si} - u_s)(P_{gi} - u_g), \tag{15}$$

where $\mu_s$, $\mu_g$, $\sigma_s$, $\sigma_g$, and $\sigma_{sg}$ represent the satellite-derived precipitation data and rainfall station data mean, variance, and covariance for each $3 \times 3$ cell, and $S$ and $G$ represent the satellite-derived precipitation products and the rainfall station data, respectively. Three constants ($c_1$, $c_2$, and $c_3$) were used to avoid instability when the denominator values of the equations were very close to zero. The values of these constants were calculated on the basis of the dynamic range of the pixel values ($R$ represents the dynamic range of the pixel values) and two small constants ($k_1 = 0.01$, $k_2 = 0.03$). Therefore, $c_1 = (k_1 R)^2$, $c_2 = (k_2 R)^2$, and $c_3 = c_2/2$. $P_{si}$ and $P_{gi}$ indicate the precipitation values for each grid in each $3 \times 3$ cell of the maps $S$ and $G$, respectively. The weight for the 9 grids in the local window, $w_i = 1/9$. The

values of SIM and SIV can vary from 0 to 1, while the value of SIP can vary from −1 to 1. The SSI is calculated as an overall measure of comparison:

$$\text{SSI}(S, G) = \text{SIM}(S, G)^{\alpha} \cdot \text{SIV}(S, G)^{\beta} \cdot \text{SIP}(S, G)^{\gamma}, \tag{16}$$

where constants $\alpha$, $\beta$, and $\gamma$ can be used to weight individual components of the SSI; in this study, $\alpha = \beta = \gamma = 1$. Therefore, the SSI is bounded by (−1, 1) where −1 indicates complete dissimilarity of the spatial structure between the satellite data and the rainfall station data, and 1 reflects identical spatial distributions of extreme precipitation. SSIM is a mean index to evaluate the overall spatial structure difference, n is the grid number.

$$\text{SSIM}(S, G) = \sum_{i=1}^{n} \text{SSI}(S_i, G_i), \tag{17}$$

## 4. Results

### 4.1. Performance of Daily Satellite-Derived Precipitation Products

#### 4.1.1. Accuracy Evaluation of Precipitation Data

The daily scale Corr, RB, AD, and RMSE of the satellite-derived precipitation products were calculated for Beijing, and the average values of the four indicators in the six subregions are shown in Table 5. It can be seen that in different subregions, the values of Corr between MSWEP and rainfall station data are significantly higher than for IMERG and CMFD. The regional averages of Corr of IMERG, MSWEP, and CMFD are 0.72, 0.81, and 0.76, respectively. Most Corr values of the three products are >0.6 over most subregions of Beijing, indicating the significant correlation between the three products and the rainfall stations. We found that the Corr between satellite-derived precipitation products and rainfall station data in mountainous areas is lower than that in urban and suburban areas, and similar trends have been found in other related studies [72,73]. This finding could be attributable to the limitations of satellite-based rainfall estimations derived from inversion of thermal infrared and passive microwave sensor retrievals in mountainous areas [74,75]. The regional averages of RB of IMERG, MSWEP, and CMFD are 3.2%, −12.4%, and 0.4%, respectively. In the UA, ISAN, NWMA, and OSA subregions, MSWEP shows obvious negative deviations, indicating that MSWEP underestimates daily precipitation in these areas. The regional averages of AD of IMERG, MSWEP, and CMFD are 1.30, 0.95, and 1.15, respectively, and those of RMSE are 4.80, 4.10, 4.49, respectively. All satellite-derived precipitation products show large AD and RMSE in the UA, NWMA, SWMA, and OSA subregions. In each subregion, the values of AD and RMSE for MSWEP are lower than those for IMERG and CMFD. From the perspective of daily precipitation accuracy, MSWEP produces smaller errors but generally underestimates daily precipitation.

**Table 5.** Accuracy evaluation of the satellite-derived precipitation data in different subregions of Beijing.

| | Index | UA | ISAS | ISAN | SWMA | NWMA | OSA | Beijing |
|---|---|---|---|---|---|---|---|---|
| Corr | IMERG | 0.74 | 0.73 | 0.76 | 0.68 | 0.70 | 0.75 | 0.72 |
| | MSWEP | 0.82 | 0.82 | 0.77 | 0.80 | 0.78 | 0.80 | 0.81 |
| | CMFD | 0.79 | 0.77 | 0.74 | 0.70 | 0.71 | 0.78 | 0.76 |
| RB | IMERG | 1.0% | 13.6% | 4.5% | 2.3% | 2.8% | −1.6% | 3.2% |
| | MSWEP | −13.0% | −2.5% | −17.9% | −6.6% | −16.0% | −26.2% | −12.5% |
| | CMFD | −5.8% | 10.0% | 4.5% | 2.6% | 2.4% | 5.4% | 0.4% |
| AD (mm) | IMERG | 1.33 | 1.29 | 1.24 | 1.27 | 1.27 | 1.32 | 1.30 |
| | MSWEP | 0.94 | 0.90 | 0.96 | 0.94 | 0.97 | 1.07 | 0.95 |
| | CMFD | 1.10 | 1.08 | 1.19 | 1.23 | 1.22 | 1.23 | 1.15 |
| RMSE (mm) | IMERG | 5.02 | 4.68 | 4.46 | 4.94 | 4.40 | 4.73 | 4.80 |
| | MSWEP | 4.22 | 3.81 | 4.30 | 4.10 | 3.81 | 4.38 | 4.10 |
| | CMFD | 4.54 | 4.37 | 4.60 | 4.76 | 4.25 | 4.41 | 4.49 |

### 4.1.2. Evaluation of Precipitation Capture Capability

This study calculated the precipitation capture capability index of the three satellite-derived precipitation products of all daily precipitation data and the daily precipitation data above the thresholds of 75th and 90th quantiles per year (Figure 2). It can be seen from the figures that the POD values of IMERG, MSWEP, and CMFD are 0.86, 0.84, and 0.93, respectively, and that the VPOD values are 0.95, 0.96, and 0.98, respectively, indicating that CMFD can correctly detect more precipitation events in all data. When precipitation at the rainfall stations is above the Q75 threshold, the POD and VPOD values of MSWEP and CMFD are very close and slightly higher than those of IMERG; however, in comparison with all data, their precipitation capture abilities are significantly reduced, as shown by values between 0.5 and 0.8. When precipitation at the rainfall stations is above Q90 threshold, the POD and VPOD values of MSWEP are the highest, but their values are <0.6. The FAR values of IMERG, MSWEP, and CMFD are 0.54, 0.33, and 0.52, respectively, and the VFAR values are 0.18, 0.09, and 0.16, respectively, indicating that MSWEP has the lowest FAR. With increase in the precipitation threshold, the FAR values of the three products do not change greatly, whereas the VFAR values increase significantly. Moreover, the CSI values of IMERG, MSWEP, and CMFD are 0.39, 0.56, and 0.44, respectively, indicating that MSWEP has the best capability for catching precipitation. The CSI values of the three products decrease with increase in the precipitation threshold. Generally, MSWEP has greater capability for capturing extreme precipitation and has a lower FAR, followed in descending order by CMFD and IMERG.

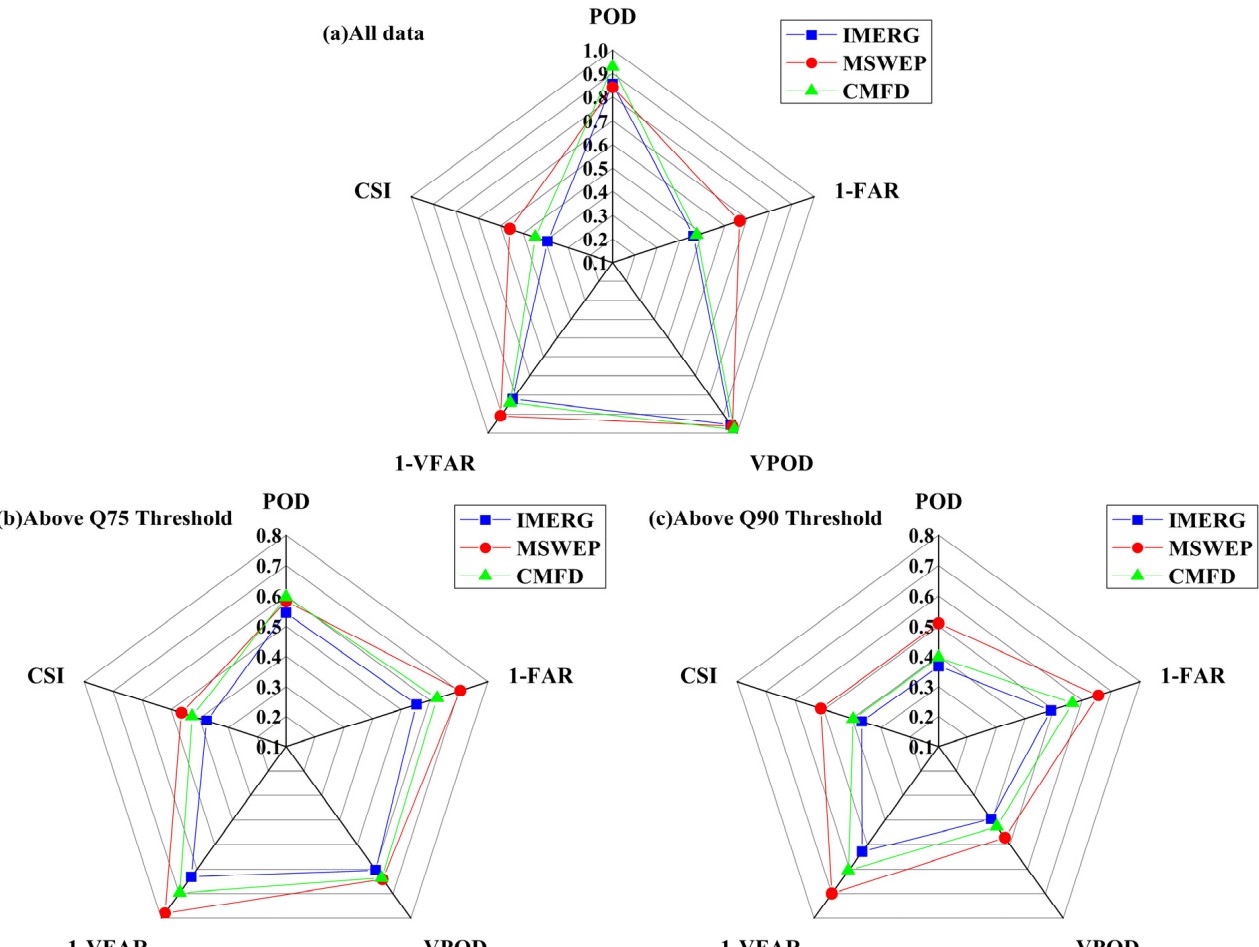

**Figure 2.** Capability of the satellite-derived precipitation products to capture precipitation with different thresholds: (**a**) all data, (**b**) above the Q75 threshold, and (**c**) above the Q90 threshold.

### 4.2. Capability for Identification of Extreme Precipitation

4.2.1. Total Indices and Persistent Indices

Figure 3 shows box plots for the rainfall station data and the three satellite-derived precipitation products for the total indices and the persistent indices (i.e., ATP, API, CDD, and CWD). The ATP values of rainfall stations, IMERG, MSWEP, and CMFD data are 522.9, 533.8, 452.1, and 520.5 mm, respectively, and the API values are 8.54, 4.67, 5.82, and 4.34 mm/d, respectively. It can be seen that the ATP values of IMERG and CMFD are very close to those of the rainfall station data, whereas there is slight underestimation by MSWEP. However, the API values for all satellite-derived precipitation products are significantly lower than those for the rainfall station data, indicating significant overestimation of the number of annual precipitation days. IMERG and CMFD reduce the differences in the API values among the different subregions of Beijing. The CDD values of the rainfall stations and the three precipitation products are 50, 30, 46, and 33 d, respectively, and the CWD values are 5, 8, 8, and 11 d, respectively. In terms of the CDD and CWD indices, all of the satellite-derived precipitation data underestimate the number of consecutive dry days and overestimate the number of consecutive wet days, with MSWEP being closest to the rainfall station data. Moreover, CMFD significantly increases the differences in CWD values among the different subregions in Beijing. It can be seen that if MSWEP data were used to evaluate precipitation in Beijing, annual precipitation would be underestimated and humidity would be underestimated, whereas use of IMERG or CMFD would result in overestimation of humidity in Beijing. Generally, MSWEP has greater accuracy in terms of the total and persistent indices in Beijing, but annual precipitation would be underestimated and the wetness degree would be underestimated if MSWEP were used to evaluate precipitation in Beijing.

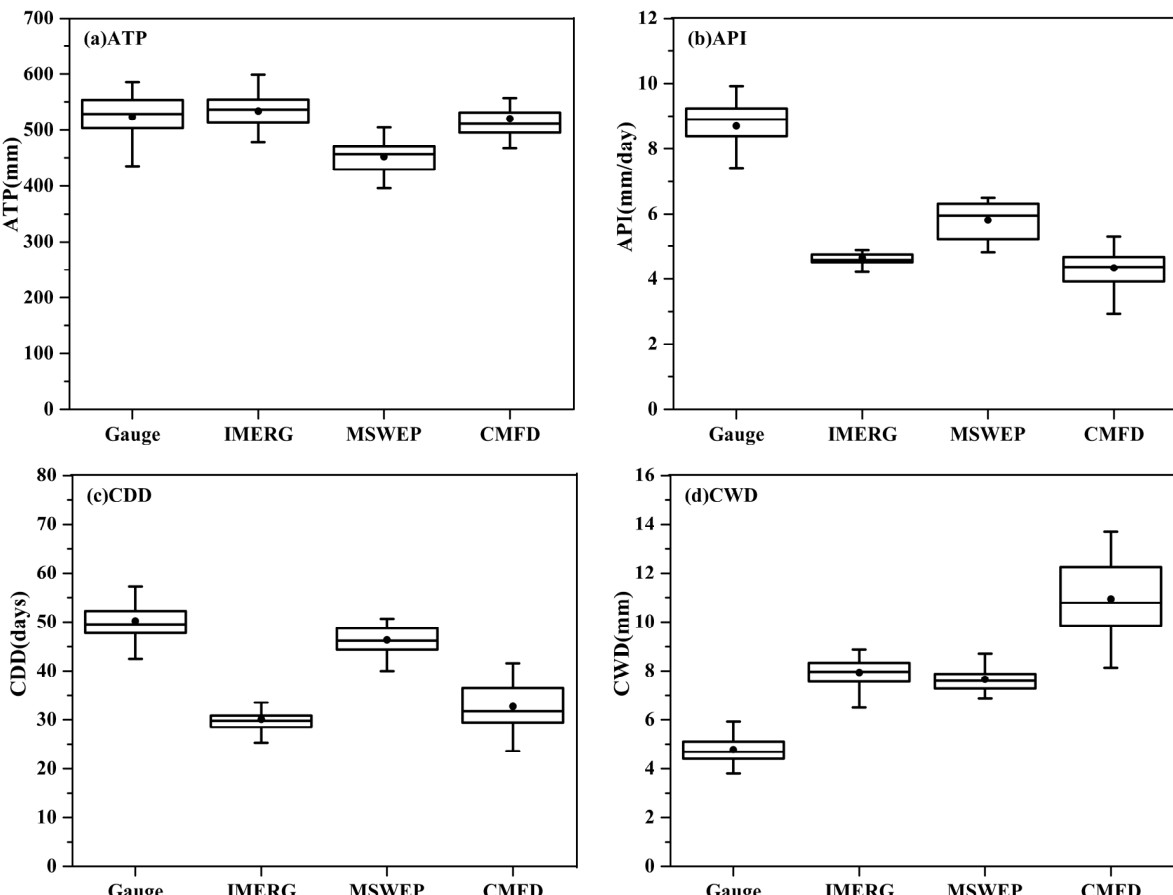

**Figure 3.** Boxplots for the four total indices and the persistent indices: (**a**) ATP, (**b**) API, (**c**) CDD, and (**d**) CWD for the rainfall station data and the three satellite-derived precipitation data products at 36 stations.

#### 4.2.2. Percentile and Max Indices

To analyze the capability of the three satellite-derived precipitation products for capturing extreme precipitation, four extreme precipitation indicators (i.e., RX1day, RX5day, R95p, and R95pTOT) of rainfall station data and IMERG, MSWEP, and CMFD data were calculated, as shown in Figures 4 and 5. It can be seen from Figure 4 that each of the four indicators of the three precipitation products has the same trend as that of the rainfall station data, but all underestimate RX1day, RX5day, and R95p. As shown in Figure 5, the RX1day values of the rainfall station data and IMERG, MSWEP, and CMFD are 76.5, 56.3, 54.7, and 58.1 mm, respectively, and the RX5day values are 108.1, 83.4, 83.9, and 89.5 mm, respectively. The three satellite-derived precipitation products have similar degrees of underestimation of the RX1day and RX5day values, but IMERG narrows the gap of the RX5day values in different subregions. The R95p values of the rainfall stations and satellite-derived precipitation products are 34.3, 20.4, 23.4, and 19.2 mm, respectively, and the R95pTOT values are 166.1, 187.9, 142.4, and 189.4 mm, respectively. The R95p values of MSWEP are closest to the values of the rainfall stations. We also found that IMERG obviously narrows the gap of the R95p values in different subregions. Overall, the three satellite-derived precipitation products have underestimated extreme precipitation indicators (i.e., RX1day, RX5day, and R95p). In terms of the R95pTOT index, the values are overestimated for IMERG and CMFD but are underestimated for MSWEP.

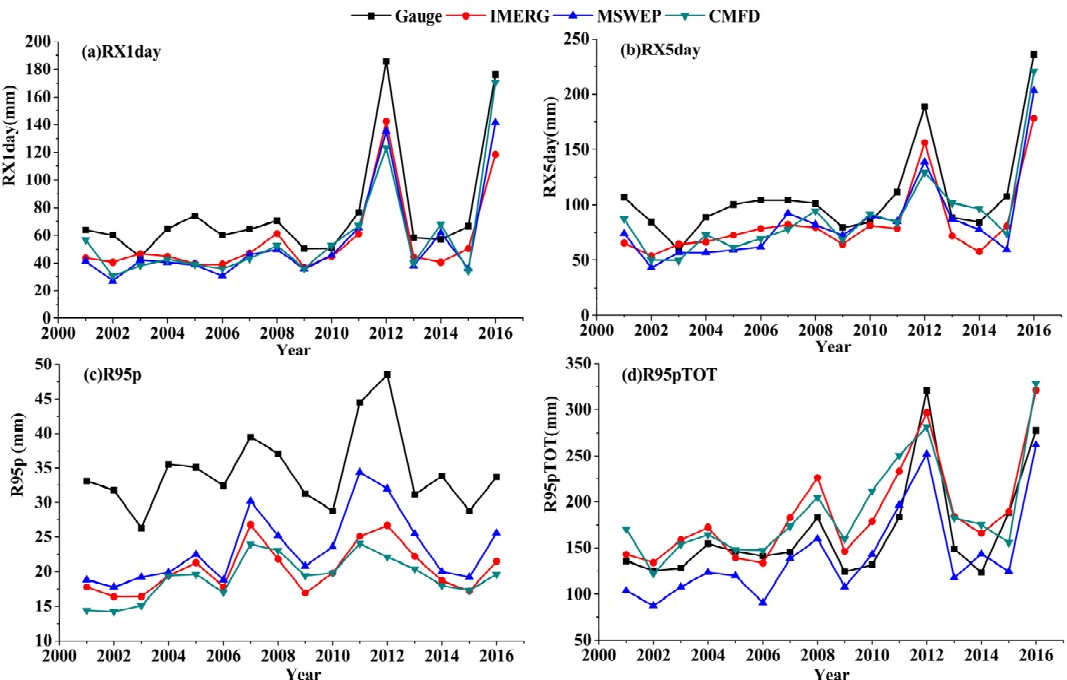

**Figure 4.** Extreme precipitation index values of the rainfall station data and satellite-derived precipitation data during 2001–2016: (**a**) RX1day, (**b**) RX5day, (**c**) R95p, and (**d**) R95pTOT.

To analyze the time consistency of extreme precipitation captured by the three satellite-derived precipitation products, we evaluated the accuracy index of the occurrence dates of RX1day detection by the rainfall station and satellite-derived precipitation products for the period 2001–2016 (Table 6). We sorted the 365 days in a year according to 1–365, calculated the serial number corresponding to the annual RX1day occurrence time of each rainfall station and the grid corresponding to the three precipitation products. It can be seen from Table 6 that the occurrence dates of RX1day detection by the three products have no significant correlation with the rainfall stations. The three products have positive deviations in the UA, SWMA, and OSA subregions, indicating that their measured occurrence dates of RX1day were later than those of the rainfall station, whereas they have advanced occurrence

dates of RX1day in the ISAS and ISAN subregions because of the negative deviation. The regional averages of AD of IMERG, MSWEP, and CMFD are 22, 22 and 25 d, respectively, and those of RMSE are 33, 35, and 38 d, respectively. IMERG and MSWEP show better performance in time consistency for the occurrence dates of RX1day, but most occurrence dates of RX1day captured by the three products remain inconsistent with the occurrence dates determined by the rainfall station data.

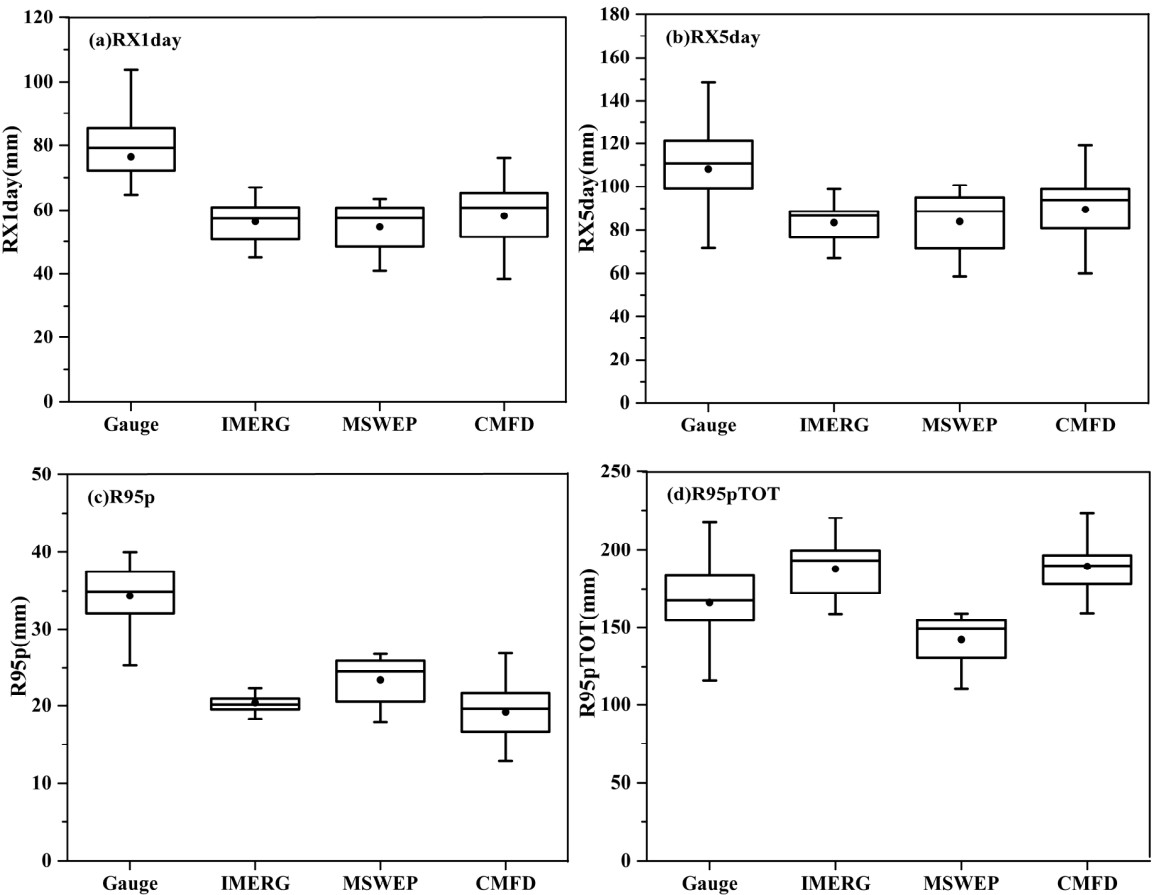

**Figure 5.** Boxplots for the four percentile and max indices: (**a**) RX1day, (**b**) RX5day, (**c**) R95p, and (**d**) R95pTOT for the rainfall station data and the three satellite-derived precipitation products at 36 stations.

**Table 6.** Accuracy evaluation of the occurrence date of RX1day detection by the rainfall station and satellite-derived precipitation products for the period 2001–2016.

| | Index | UA | ISAS | ISAN | SWMA | NWMA | OSA | Beijing |
|---|---|---|---|---|---|---|---|---|
| Corr | IMERG | 0.41 | 0.41 | 0.22 | 0.18 | 0.48 | 0.26 | 0.36 |
| | MSWEP | 0.48 | 0.27 | 0.40 | 0.19 | 0.29 | 0.45 | 0.37 |
| | CMFD | 0.17 | 0.39 | 0.47 | 0.11 | 0.31 | 0.19 | 0.23 |
| RB | IMERG | 1.7% | −1.6% | −1.4% | 4.4% | −0.7% | 1.4% | 1.0% |
| | MSWEP | 4.3% | −1.9% | −1.3% | 6.3% | 2.4% | 2.1% | 2.9% |
| | CMFD | 0.8% | −1.7% | −1.8% | 3.3% | −1.2% | 0.4% | 0.3% |
| AD (days) | IMERG | 20 | 25 | 23 | 28 | 22 | 19 | 22 |
| | MSWEP | 18 | 24 | 17 | 29 | 28 | 19 | 22 |
| | CMFD | 23 | 25 | 18 | 32 | 27 | 19 | 25 |
| RMSE (days) | IMERG | 31 | 38 | 34 | 40 | 32 | 29 | 33 |
| | MSWEP | 29 | 43 | 27 | 45 | 41 | 30 | 35 |
| | CMFD | 36 | 39 | 28 | 50 | 42 | 31 | 38 |

### 4.2.3. Absolute Threshold Indices

According to the classification of precipitation grades by the meteorological department and the characteristics of precipitation in Beijing, daily precipitation grades are divided into the following five categories: light precipitation (0.1–5.0 mm/d), moderate precipitation (5.1–10 mm/d), heavy precipitation (10.1–25.0 mm/d), torrential precipitation (25.1–50.0 mm/d), and extraordinary rainstorm (>50 mm/d) [48]. By calculating the AD and RB of the precipitation amount and the precipitation days produced by satellite-derived precipitation products at different precipitation levels (Figure 6), it was found that all satellite products mainly overestimate the amount of precipitation and the number of precipitation days for slight and moderate precipitation, and mainly underestimate the amount of precipitation and the number of precipitation days when the precipitation level is >10 mm/d. When the precipitation level is <50 mm, the AD and RB values of the precipitation produced by MSWEP are relatively small, especially for light precipitation, e.g., the RB of precipitation produced by MSWEP is 50%, which is much lower than the 101% produced by IMERG and CMFD. Moreover, the AD and RB values of precipitation days produced by MSWEP at this level are the lowest. Generally, IMERG, MSWEP, and CMFD significantly overestimate precipitation and the number of precipitation days for light precipitation, and significantly underestimate precipitation and the number of precipitation days for torrential precipitation and extraordinary rainstorms. When daily precipitation is <50 mm/d, MSWEP performs best, and when daily precipitation is >50 mm/d, CMFD performs best.

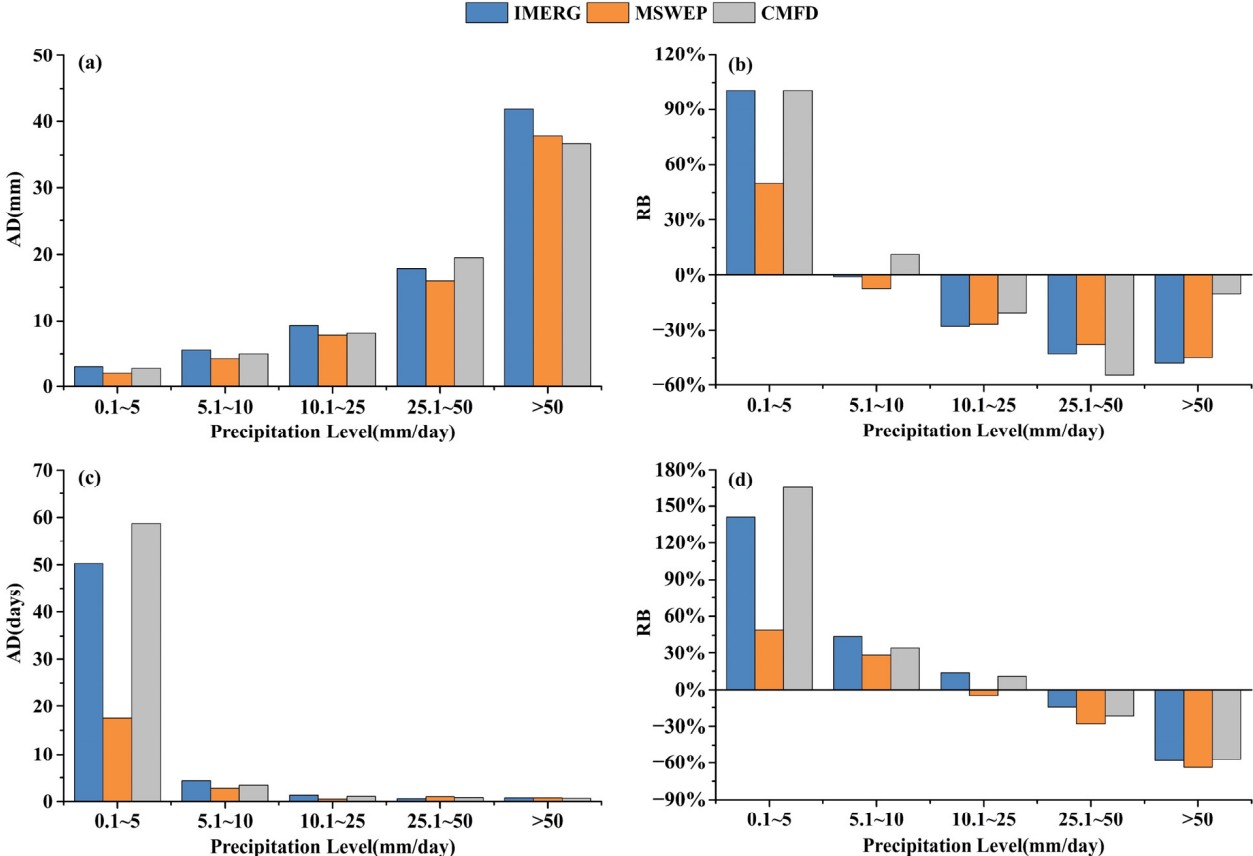

**Figure 6.** Absolute and relative deviations between the satellite-derived precipitation data and rainfall station data for different precipitation levels: (**a**) AD of precipitation, (**b**) RB of precipitation, (**c**) AD of number of precipitation days, and (**d**) RB of number of precipitation days.

### 4.3. Spatial Comparison Statistics of Extreme Rainstorms

Extreme rainstorms occurred in Beijing on 21 July 2012 ("7.21") and 20 July 2016 ("7.20"), which triggered secondary disasters such as urban waterlogging and mountain torrents, causing massive loss of life and substantial damage to property. We calculated the SIM, SIV, SIP, and SSI values of the two extreme rainstorms captured by the three satellite-derived precipitation products relative to the rainfall stations (Figures 7–10). It can be seen that local means of the three satellite products are comparable with the local means of the rainfall stations for the "7.21" rainstorm in 2012 (SIM is close to 1), but they show a slightly lower SIM value in the southwest of Beijing. MSWEP and CMFD performed well for the "7.20" rainstorm in 2016, but IMERG shows some discrepancy relative to the rainfall stations over the SWMA subregion. In Figure 8, MSWEP displays larger differences in local variance over Beijing for the two rainstorms, and CMFD performed best with its SIV value close to 1 over the region. SIP represents the spatial correlation between satellite precipitation products and rainfall stations, and SIP values close to 1 indicate similarity in locations of high and low variance between the satellite products and the rainfall stations. Comparison of Figure 9a–f reveals that the three precipitation products all show negative values in plain areas, especially the UA and ISAS subregions, and positive values in mountainous areas for the "7.21" rainstorm in 2012; IMERG shows relatively low correlation with rainfall stations throughout the entire region. From the perspective of the comprehensive SSI (Figure 10a–f), the SSI values of the two rainstorms are generally affected by the SIP values, indicating that SIP is the main component contributing to the inconsistency in the spatial structure between the three precipitation products and the rainfall stations over Beijing. They show obvious spatial structure inconsistencies in the plain areas (mainly urban and southern suburbs) for the "7.21" rainstorm in 2012. We calculated the SSIM of IMERG, MSWEP, and CMFD, which were 0.44, 0.59, and 0.62 in 2012 and 0.34, 0.49, and 0.53 in 2016, respectively. Generally, MSWEP and CMFD performed better than IMERG.

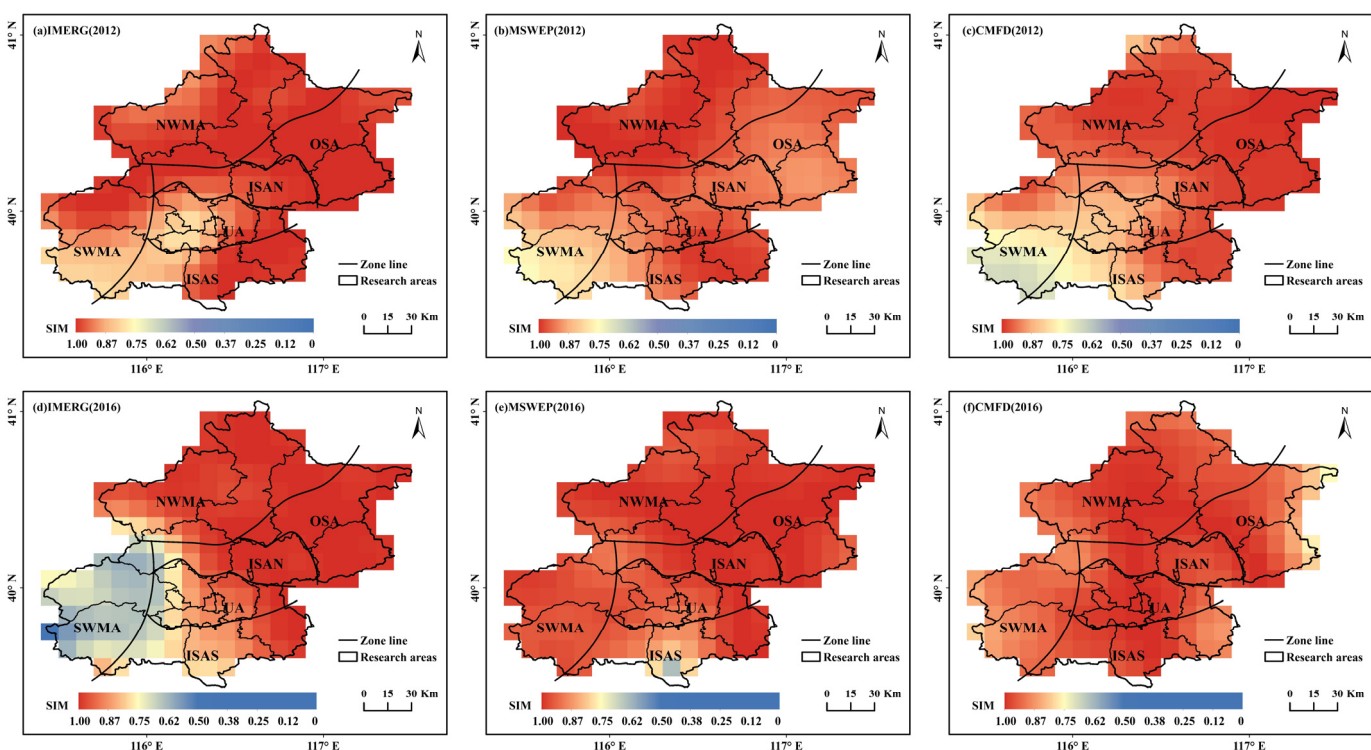

**Figure 7.** Similarity in mean index (SIM) of the extreme rainstorms that occurred in Beijing on 21 July 2012 and 20 July 2016: (**a**) IMERG (2012), (**b**) MSWEP (2012), (**c**) CMFD (2012), (**d**) IMERG (2016), (**e**) MSWEP (2016), and (**f**) CMFD (2016).

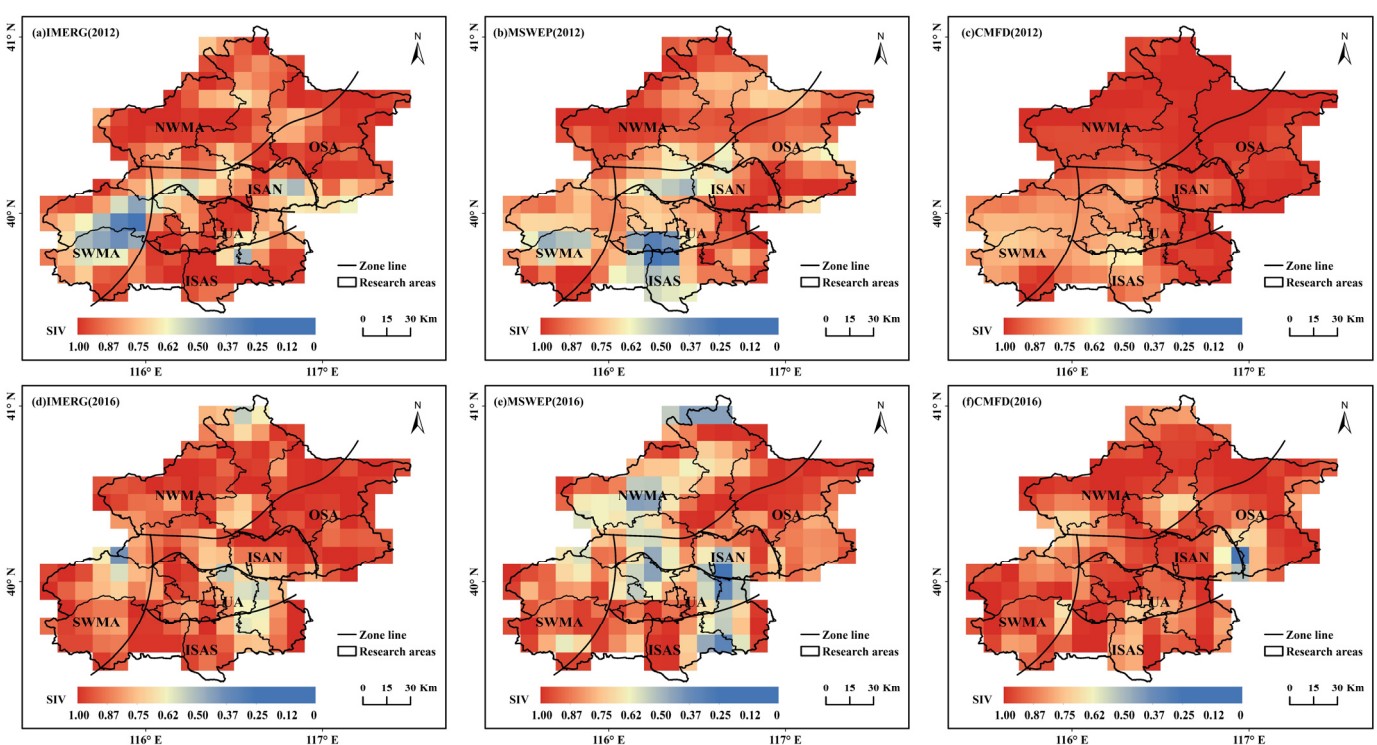

**Figure 8.** Similarity in variance index (SIV) of the extreme rainstorms that occurred in Beijing on 21 July 2012 and 20 July 2016: (**a**) IMERG (2012), (**b**) MSWEP (2012), (**c**) CMFD (2012), (**d**) IMERG (2016), (**e**) MSWEP (2016), and (**f**) CMFD (2016).

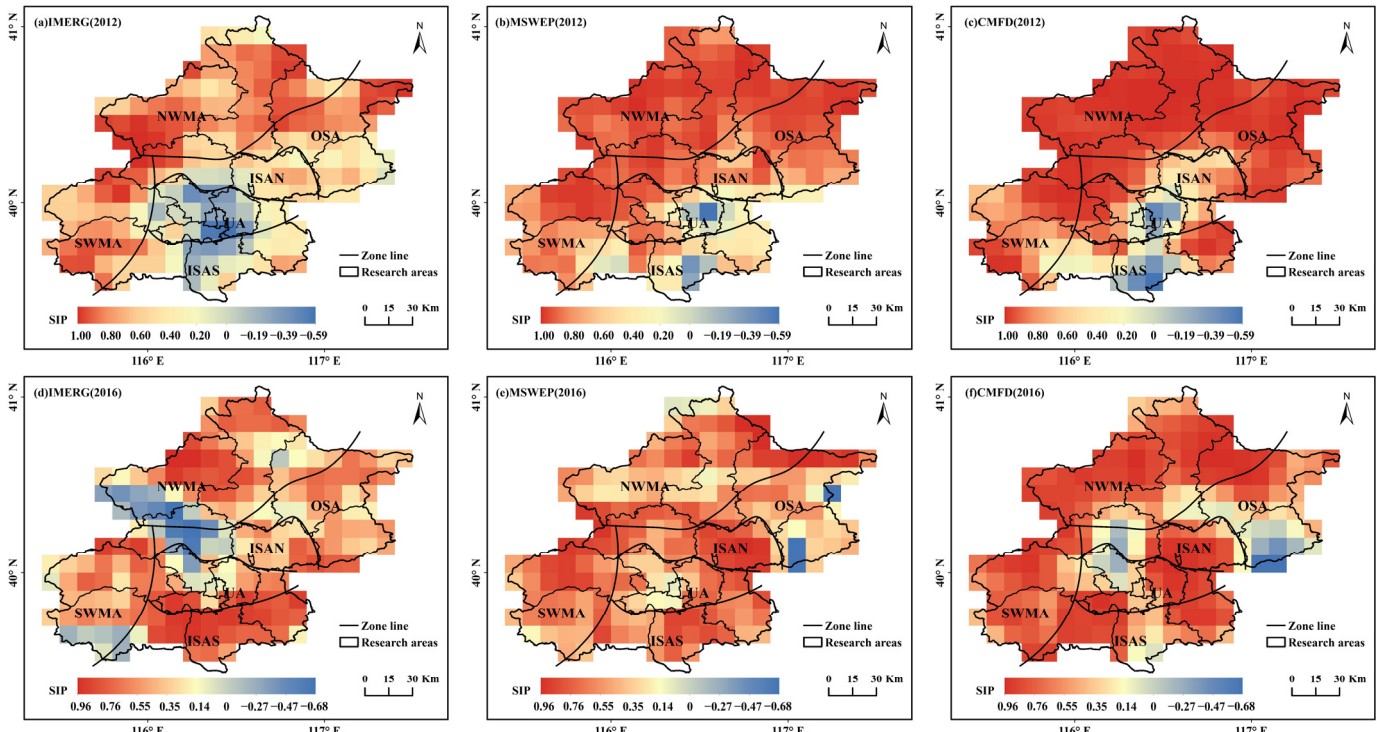

**Figure 9.** Similarity in pattern (SIP) of the extreme rainstorms that occurred in Beijing on 21 July 2012 and 20 July 2016: (**a**) IMERG (2012), (**b**) MSWEP (2012), (**c**) CMFD (2012), (**d**) IMERG (2016), (**e**) MSWEP (2016), and (**f**) CMFD (2016).

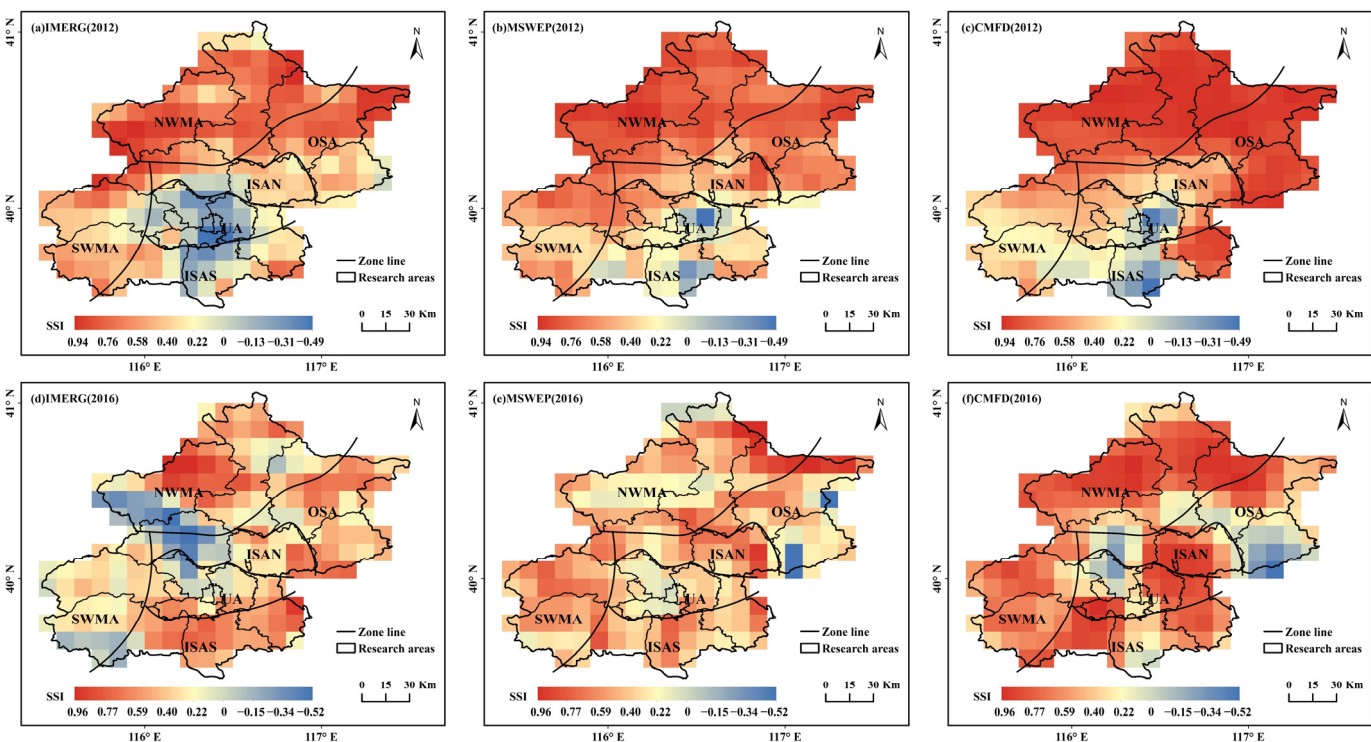

**Figure 10.** Structural Similarity Index (SSI) of the extreme rainstorms that occurred in Beijing on 21 July 2012 and 20 July 2016: (**a**) IMERG (2012), (**b**) MSWEP (2012), (**c**) CMFD (2012), (**d**) IMERG (2016), (**e**) MSWEP (2016), and (**f**) CMFD (2016).

## 5. Discussion

This study assessed the accuracy and capability of IMERG, MSWEP, and CMFD in capturing daily precipitation and extreme precipitation events in Beijing in relation to rainfall station data. We selected daily precipitation data from 2001–2016 and found that MSWEP has better performance than both IMERG and CMFD. MSWEP has higher accuracy in terms of daily precipitation estimation with the highest Corr and lowest AD values. However, MSWEP mainly underestimates daily precipitation in the different subregions of Beijing, resulting in a relatively large RB, while IMERG and CMFD mainly overestimate daily precipitation. CMFD has the greatest capability for capturing precipitation, which might be attributable to the integration of conventional meteorological observations from the China Meteorological Administration. MSWEP has the lowest percentage of incorrectly captured precipitation events. Generally, MSWEP has greater capability for capturing extreme precipitation and has a lower FAR, followed in descending order by CMFD and IMERG.

By comparing and analyzing 13 extreme precipitation indices of the three satellite-derived precipitation products and rainfall station observations, we found that IMERG, MSWEP, and CMFD underestimate CDD and overestimate CWD. The three products would overestimate the degree of humidity when assessing precipitation in Beijing owing to overestimation of the number of annual precipitation days. It can be seen that MSWEP has higher accuracy in terms of the total and persistent indices in relation to Beijing. All three products underestimate the RX1day, RX5day, and R95p indices. Both IMERG and CMFD mainly overestimate R95pTOT, whereas it is mainly underestimated by MSWEP. IMERG and MSWEP show better performance in time consistency for the occurrence dates of RX1day, but the occurrence dates of RX1day captured by the three products remain inconsistent with that derived from rainfall station observations because their AD is >20 d. On the basis of the performance regarding different precipitation levels, IMERG, MSWEP, and CMFD significantly overestimate precipitation and the number of precipitation days

for light precipitation, and significantly underestimate precipitation and the number of precipitation days for torrential precipitation and extraordinary rainstorms, which is a finding in accordance with the results for Beijing reported by Ren [76].

Many studies [76,77] have shown that topography (mainly elevation) affects the accuracy of satellite-derived precipitation products; however, it can be seen from Table 5 that the values of AD, RB, and RMSE associated with the three satellite-derived precipitation products in mountainous and plain areas have no obvious regional differences. Moreover, as MSWEP and CMFD are precipitation fusion products integrated with station precipitation data, whereas IMERG is solely a satellite-derived precipitation product, the accuracy and precipitation capture capability of IMERG are not as good as those of MSWEP and CMFD. However, in future research, precipitation fusion products and satellite-derived precipitation products should be compared separately to better reflect the performance of different satellite-derived precipitation products. Additionally, the commonly used satellite precipitation products with a high spatial resolution of 0.1 degree is still a little coarse for Beijing. Moreover, due to the uneven distribution of rainfall stations in Beijing, dense stations in urban areas and sparse stations in mountainous areas, the grid matching accuracy and comparison accuracy will be affected. At present, the rainfall stations in Beijing are under construction and improvement. We hope to use the rainfall station data and the satellite-derived precipitation data with higher spatial-temporal resolution to evaluate extreme precipitation events at shorter durations in future work.

## 6. Conclusions

In this study, we used observations from 36 rainfall stations in Beijing as reference data to evaluate the applicability of the three satellite-derived precipitation products: IMERG_V06, MSWEP V2, and CMFD. In addition to detecting their capability in capturing the frequency, intensity, and occurrence date of precipitation, we also used the novel SSI spatial map comparison method to compare the spatial distributions of two extreme rainstorms that occurred in Beijing.

We found that MSWEP had the highest Corr of daily precipitation with rainfall station data, and the lowest AD and RMSE values; however, it presented systematic underestimation. CMFD had a higher detection rate, but MSWEP always maintained the lowest FAR for all data and for data above the 75% and 90% thresholds. In terms of the monitoring of extreme precipitation indicators, the three products all underestimated RX1day, RX5day, and R95p. The R95pTOT indicator was overestimated by IMERG and CMFD but underestimated by MSWEP. Moreover, MSWEP showed higher accuracy in estimating precipitation amount and the number of precipitation days for different precipitation levels. Of the three satellite-derived products, the SSI evaluation showed that MSWEP and CMFD have greatest similarity to the two extreme rainstorms, i.e., "7.21" in 2012 and "7.20" in 2016. Overall, of the three products, MSWEP was found most suitable for application to the Beijing region, but it produced systematic underestimation and should be used with caution in relation to early warning of urban flood disasters.

**Author Contributions:** Conceptualization, Y.L. and B.P.; Data curation, Y.L. and B.P.; Formal analysis, Y.L.; Software, Y.L.; Validation, S.S., D.P. and Z.Z.; Writing—Original draft preparation, Y.L.; Writing—Review and Editing, B.P., M.R. and D.Z. All authors have read and agreed to the published version of the manuscript.

**Funding:** This research was funded by The National Natural Science Foundation of China (51879008), The National Natural Science Foundation of China (52179003).

**Data Availability Statement:** IMERG V06 dataset is publicly available (at https://gpm.nasa.gov/data/directory, accessed on 13 April 2022); MSWEP V2 dataset is publicly available (at http://www.gloh2o.org/mswep, accessed on 13 April 2022); CMFD dataset is publicly available (at http://data.tpdc.ac.cn/zh-hans/data/8028b944-daaa-4511-8769-965612652c49/, accessed on 13 April 2022). The rainfall stations data used in this study is provided by the Beijing Hydrology Bureau and the National Climate Center of China and the data are not publicly available due to privacy policy.

**Acknowledgments:** The authors thank the Beijing Hydrology Bureau and the National Climate Center of China for providing ground observation data.

**Conflicts of Interest:** The authors declare no conflict of interest.

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
