# Peer review of "Evaluation of Performance of Three Satellite-Derived Precipitation Products in Capturing Extreme Precipitation Events over Beijing, China"

_remotesensing, doi:10.3390/rs14112698_

Round 1

Reviewer 1 Report

Please find the comments as attached.

Reviewer 2 Report

This paper explored performance of three satellite-derived precipitation products in capturing extreme precipitation events over Beijing, china. The topic is interesting and has implications in extreme precipitation events over high-density urban agglomeration regions. The paper is well organized and written. The findings of this study are worth of publication in the journal after minor revision as follows:

1) When comparing extreme precipitation indexes, are the time ranges selected by these precipitation samples consistent? This will affect the comparability of 13 extreme precipitation indexes. Please explain clearly.

2) In addition, when matching the site data, is the point to pixel approach the closest match? The author should give a simple explanation. It may be difficult for readers to have the conditions or time to consult the literature to understand this method.

3) The precipitation data products of MSWEP and CMFD have integrated the rain gauge data. If they are compared with the site data, the overall effect should be better. It is suggested to have a brief discussion.

4) As far as Beijing is concerned, the resolution of 0.1 degree is still a little coarse. The distribution density of rain gauge determines the grid matching accuracy and comparison accuracy. It is suggested to discuss this issue.
